:ۉ: PLOS | ONE

# 2'-O-ribose methylation of transfer RNA promotes recovery from oxidative stress in *Saccharomyces cerevisiae*

**Lauren Endres**[ORCID][1]*, **Rebecca E. Rose**[2], **Frank Doyle**[3], **Taylor Rahn**[1], **Bethany Lee**[1], **Jessica Seaman**[1], **William D. McIntyre**[4], **Daniele Fabris**[4]

**1** College of Arts and Sciences, SUNY Polytechnic Institute, Utica, New York, United States of America, **2** Metabolomics Core Resource Laboratory, New York University Langone Heath, New York, New York, United States of America, **3** College of Nanoscale Science and Engineering, SUNY Polytechnic Institute, Albany, New York, United States of America, **4** Department of Chemistry, University of Connecticut, Storrs, Connecticut, United States of America

* endresl@sunypoly.edu

**Data Availability Statement:** All relevant data are within the paper and its Supporting Information files.

## Abstract

Chemical modifications that regulate protein expression at the translational level are emerging as vital components of the cellular stress response. Transfer RNAs (tRNAs) are significant targets for methyl-based modifications, which are catalyzed by tRNA methyl-transferases (Trms). Here, *Saccharomyces cerevisiae* served as a model eukaryote system to investigate the role of 2'-O-ribose tRNA methylation in the cell's response to oxidative stress. Using 2'-O-ribose deletion mutants for *trms* 3, 7, 13, and 44, in acute and chronic exposure settings, we demonstrate a broad cell sensitivity to oxidative stress-inducing toxicants (i.e., hydrogen peroxide, rotenone, and acetic acid). A global analysis of hydrogen peroxide-induced tRNA modifications shows a complex profile of decreased, or undetectable, 2'-O-ribose modification events in 2'-O-ribose *trm* mutant strains, providing a critical link between this type of modification event and Trm status post-exposure. Based on the pronounced oxidative stress sensitivity observed for *trm7* mutants, we used a bioinformatic tool to identify transcripts as candidates for regulation by Trm7-catalyzed modifications (i.e., enriched in UUC codons decoded by tRNA$^{PheGmAA}$). This screen identified transcripts linked to diverse biological processes that promote cellular recovery after oxidative stress exposure, including DNA repair, chromatin remodeling, and nutrient acquisition (i.e., *CRT10*, *HIR3*, *HXT2*, and *GNP1*); moreover, these mutants were also oxidative stress-sensitive. Together, these results solidify a role for *TRM3*, *7*, *13*, and *44*, in the cellular response to oxidative stress, and implicate 2'-O-ribose tRNA modification as an epitranscriptomic strategy for oxidative stress recovery.

## Introduction

Transfer RNAs are fundamental biological molecules that read cognate codons in mRNA to complete the flow of genetic information from DNA to protein. A ubiquitous feature of tRNAs

**Funding:** This research was completed with support from a SUNY Polytechnic Institute Seed Grant (LE, https://sunypoly.edu/sponsored-research.html) and an award from the Dr. Nuala McGann Drescher Diversity and Inclusion Leave Program (LE, https://goer.ny.gov).

**Competing interests:** The authors have declared that no competing interests exist.

is chemical modification, the most prominent being methylation (of the base or ribose sugar) as catalyzed by tRNA methyltransferase enzymes (Trms)[1, 2]. Notably, tRNA methylation occurs post-transcriptionally, and can thus be viewed as an epitranscriptomic feature at the molecular level [3]. In general, methyl-based modifications of tRNA function to ensure translational fidelity, although how cells accomplish this depends upon the chemical nature of the modification and its position within the tRNA molecule [4]. For example, modifications within the anticodon stem loop (ASL) at position 34 (decoding the wobble base) and 37 influence both the accuracy and efficiency of translation through degenerate codon selection [5–7]. Modifications within other loop regions of tRNA can influence base interactions that direct folding, increase stability, or alter the structure of the tRNA molecule in ways that regulate aminoacylation or interactions with the ribosome [8–12].

This study is motivated by two observations that support a role for tRNA modification systems (i.e., Trm enzymes, their modifications, and target transcripts) in the oxidative stress response. First are studies in several eukaryotic cells demonstrating that deleting Trms results in sensitivity to one or more toxicants that induce oxidative stress, or damage DNA [13–16]. Second are observations that tRNA modifications are dynamic, changing in response to various toxicants that induce oxidative stress; moreover, these changes include methylation of the ribose sugar moiety at the 2'-OH position (i.e., 2'-O-ribose methylation) [17, 18]. Collectively, these observations are a good indication that Trms and their corresponding modifications play a role in the cellular response to stress. However, questions remain about how cells engage these tRNA modification systems to survive and recover from oxidative stress after toxicant exposure.

Here we focused on *Saccharomyces cerevisiae* (budding yeast) 2'-O-ribose methyltransferases because 2'-O-ribose modified bases are known to increase under conditions of oxidative stress. For example, 2'-O-methylcytosine (Cm), 2'-O-methyladenosine (Am), and 5-methoxycarbonylmethyl-2'-O-methyluridine (mcm$^5$Um)[16, 17], and unpublished observations). Furthermore, methylation of tRNA at the 2'-OH position of the ribose sugar is generally thought to increase the stability of tRNA via mechanisms that protect against spontaneous hydrolysis or nuclease digestion (e.g., in non-helical regions) and reinforce intra-loop interactions that stabilize the tertiary structure of the molecule [11, 19]]. Such stabilization strategies are essential for thermophilic organisms where tRNAs must carry out protein translation at denaturing temperatures, an idea supported by observations in diverse archaea that show an abundance of 2'-O-methylated nucleosides among global populations of modified tRNAs [20–23]. In mesophilic prokaryotes and eukaryotes, 2'-O-ribose modification also plays a structural role in the non-duplex loop regions of the D- and T-arm [11, 24, 25]. The importance of this type of methylation event in mesophilic eukaryotes is not fully understood, though it seems plausible that it serves a stabilizing function in environments that are destabilizing to tRNA molecules (e.g., increased intracellular reactive oxygen species, or ROS).

*S. cerevisiae* tRNAs undergo methylation of the 2'-OH position of the ribose sugar at three sites in the body of the tRNA molecule (i.e., 4, 18, and 44), and at two sites within the ASL (i.e., 32 and 34, the latter decoding the wobble base)[26]. At least four Trms are known to carry out these methylation events: Trm3 → Gm18, Trm7 → Cm32 and Nm34, Trm13 → Cm4, and Trm44 → Um44 (Fig 1). To investigate a role for 2'-O-ribose methylation of tRNA in the oxidative stress response, we assessed corresponding *trm* deletion mutants for their sensitivity to oxidative stress after exposure to hydrogen peroxide, rotenone, and acetic acid. In all assays, hydrogen peroxide was the most potent inducer of oxidative stress sensitivity. Accordingly, we characterized methylation chemistries in 2'-O-ribose mutants after acute exposure to hydrogen peroxide and found a complex profile of methylation events in exposed cells, which included increases in Um, Cm, and Gm in wild type cells. Notably, 2'-O-ribose associated

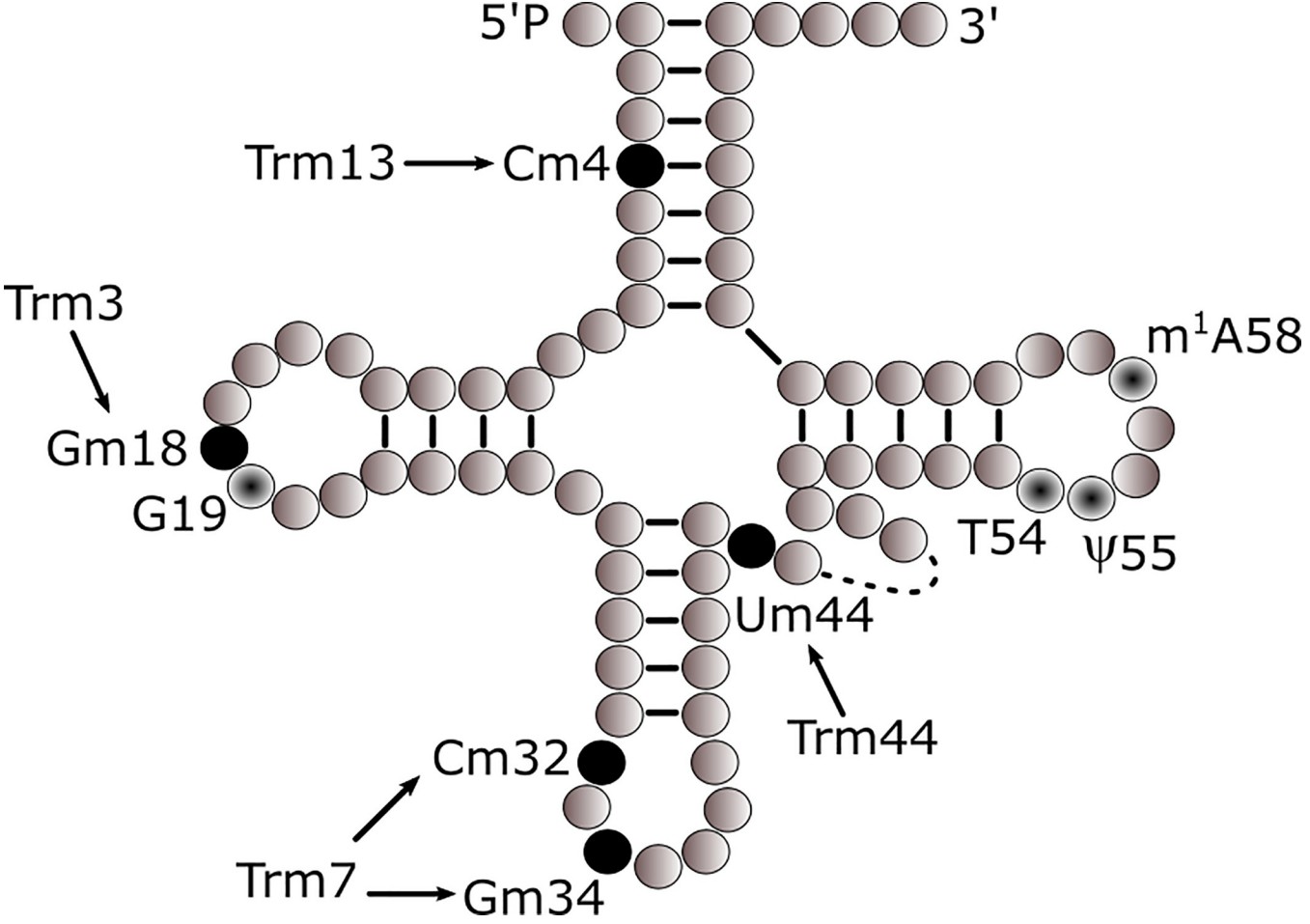

**Fig 1. 2'-O-methylation in *Saccharomyces cerevisiae*.** 2'-O-ribose methyltransferases and target nucleotides (black-filled) in a two-dimensional cloverleaf tRNA structure based on the conventional numbering system (Machnicka, Olchowik et al. 2014). Also shown are other commonly modified nucleotides in tRNAs (dot-filled).

modification chemistries were attenuated in *trm3*, *13*, and *44* mutants. Potential roles for these modifications in the oxidative stress response are discussed further in the context of their position on the tRNA molecule.

To test the idea that tRNA modification systems targeting the ASL regulate the oxidative stress response, we used Trm7 → tRNAPhe$^{(GmAA)}$ as a model for preferred UUC codon selection [27–29]. Using a bioinformatics approach [30], we identified UUC-enriched transcripts in *S. cerevisiae* as potential Trm7-influenced translational targets and found several with biological functions linked to the oxidative stress response. *GNP1* stood out as a UUC-enriched transcript with significant oxidative stress-sensitivity in corresponding *gnp1* mutants. In light of recent evidence that *trm7* mutants activate the general amino acid control (GAAC) response pathway [31], we propose that attenuated codon-biased translation of *GNP1* is a component of this response, contributing to the oxidative stress sensitivity phenotype of the *trm7* mutants, which is a focus for future study. Overall, this work solidifies a role for 2'-O-ribose methylation in the oxidative stress response and supports a model in which *S. cerevisiae* uses this type of modification event in an epitranscriptomic strategy that combines increased tRNA stability and enhanced translational effects post-exposure.

## Materials and methods

### *S. cerevisiae* cell culture and transformation

Deletion mutants were retrieved from an *S. cerevisiae* library (Open Biosystems, Thermo Scientific, Hudson, NH) established by the *Saccharomyces* Genome Deletion Project and are on a BY4741 background (i.e., haploid genotype: *MATa his3Δ1 leu2Δ0 met15Δ0 ura3Δ0*)[32]. In this library, ORFs have been replaced by a KanMX cassette, selected for by growth in yeast extract-peptone-dextrose (YPD) broth supplemented with G418 (200 μg/mL) prior to toxicant exposure.

Open reading frames (ORFs) for *TRM3*, *7*, and *13* were obtained in the galactose-inducible BG1805 plasmid vector from the Yeast ORF Collection (Dharmacon™, Horizon Discovery Group, Cambridge, UK)[33]. ORF plasmids were introduced into their respective *trm* mutant strains by transformation using a one-step buffer containing 40% polyethylene glycol (PEG), 200 mM lithium acetate and 100 mM dimethyl sulphoxide [34]. *TRM*-ORF transformants were selected by plating on synthetic complete medium: yeast nitrogen base/complete amino acid supplement mixture (YNB/CSM) minus uracil, plus 2% dextrose (SC-Ura, Sunrise Science Products, TN) and growth at 30˚C for three days.

For ORF induction, single overnight colonies were initially grown in SC-Ura, and then diluted 1:100 in YNB/CSM minus uracil plus 2% raffinose (ACROS) and allowed to reach stationary phase. These cultures were then diluted three-fold with concentrated media to either induce or repress ORF expression. That is, 3 x YNB/CSM-Ura plus 6% galactose (SIGMA) or 6% dextrose (SIGMA), respectively. After growth to an $OD_{600nm}$ of 2.0, cultures were split into separate tubes for one-hour control (water) or hydrogen peroxide (20 mM) treatments at 30˚C, followed by plating on SC-Ura (i.e., 100 μL of a $10^{-5}$ dilution) and colony counting after three days of growth at 30˚C.

### Growth, viability, and colony-forming assays

For the plate-based growth assays, individual mutants were inoculated in liquid, incubated overnight at 30˚C with shaking (i.e., 225 rpm), and then diluted in a ten-fold series (i.e., $10^{-1}$ to $10^{-5}$). Five μL of each dilution was then plated on either YPD, or YPD supplemented with hydrogen peroxide (2 mM), rotenone (400 μM), and acetic acid (20 mM). Plates were incubated at 30˚C for 3 days and imaged for growth using a digital camera system (Bio-Rad, CA). Also, quantitative measurements of growth over a twenty hour period were performed on overnight cultures grown from single colonies using an automated microplate reader set at 30˚C with orbital shaking (TECAN). To assess viability post-exposure, cultures of *trm* mutants were grown to log-phase from single colonies in YPD + G418, and then diluted to 3 x $10^6$ cells/ mL in YPD or YPD supplemented hydrogen peroxide (40 mM), rotenone (400 mM), and acetic acid (80 mM); BY4741 were similarly treated and served as the control strain. The number of viable cells was determined by staining with trypan blue, four and eight hours post-exposure; cell death was determined by dividing the number of trypan blue positive cells by the total number of cells and is calculated as a percentage.

For the colony-forming assay, similar liquid cultures of mutant single colonies and BY4741 were grown to stationary phase in rich media (i.e., YPD) and diluted to a uniform $OD_{600nm}$ of 2 (~ 6 x $10^7$ cells/mL). Hydrogen peroxide (10 mM), rotenone (100 μM) or acetic acid (20 mM) were added directly to 1 mL of these cultures, with an equivalent volume of water serving as a control. These cultures were incubated for 1 hour at 30˚C with shaking, serially diluted to $10^{-5}$, and plated on YPD (100 μL). After 3 days of growth at 30˚C, the number of colonies was counted. Similar colony forming assays were performed on *trm* mutants transformed with

their respective ORFs, albeit with different growth conditions for ORF selection and expression as described above.

## Quantitative real-time reverse-transcription PCR (qRT-PCR)

Total RNA was first isolated from BY4741, and *trm3*, *7*, *13*, and *44* mutant strains using TRIzol® reagent (ThermoFisher Scientific) and then 1 µg was converted into complementary DNA (cDNA) using an iScript™ cDNA synthesis kit (Bio-Rad) according to the manufacturer's instructions. cDNAs were diluted 5 fold and 4 µL was used to detect *TRM3*, *7*, *13*, and *44* gene expression with TaqMan® gene-specific expression assays via qRT-PCR on a StepOnePlus real-time PCR system (ThermoFisher Scientific). *TRM* expression in the *trm* mutants was determined relative to BY4741 using the delta-delta cycle threshold method ($2^{-\Delta\Delta Ct}$), with actin serving as the internal control.

## Transfer RNA modification analysis

Transfer RNA was isolated from yeast spheroplasts generated by digestion with lyticase (from *Arthrobacter luteus*, Sigma-Aldrich, MO) digestion [35] using a micro RNeasy RNA isolation kit, modified to prevent the loss of covalent modification chemistries. Specifically, at the first step of cell lysis, QIAzol reagent was supplemented with the deaminase inhibitor tertrahydrouridine (0.2 mM, Millipore Sigma, MO), and anti-oxidants deferoxamine and butylhydroxytoluene (both at 0.1 mM, Millipore Sigma, MO). At least 50 µg of total micro RNA was isolated by this method (containing > 85% tRNAs) for subsequent mass spectrometric (MS) analysis of modification chemistries.

The MS analysis was performed on tRNAs isolated from *trm* mutant strains, before and after acute exposure to hydrogen peroxide (20 mM for 1 hour), as previously described in detail, and for distinct cellular stress responses [36–38]. Briefly, isolated tRNAs were digested with select nucleases to obtain analogous mononucleotide mixtures for analysis by direct infusion electrospray ionization (ESI) to detect variable levels of expression in individual modification chemistries. Expression levels were calculated using the Abundance versus Proxy method (AvP), which determines the relative abundance of each post-translational modification (PTM) normalized against the sum of the intensities of the four canonical bases. Statistically significant differences in AvP values (exposed versus unexposed) were determined using an unpaired Student *t*-test (n = 3), and those with a p-value < 0.05 are shown in Fig 4 and appear in color in S1 and S2 Tables. The same tRNA isolates were also digested with a combination of select nucleases and phosphatases to obtain individual mononucleotides for Tandem MS analysis, carried out in both the positive and negative modes. This approach facilitated the differentiation of isobars detected during PTM analysis, including Am, Um, Cm, and Gm [36, 38].

## Intracellular ROS detection

*S. cerevisiae trm* mutants and BY4741 strains were inoculated into YPD from single colonies. When the culture was at mid-log phase ($OD_{600nm} \sim 0.5$), cells were pelleted by centrifugation (2500 rpm for 5 min), washed once with phosphate-buffered saline (PBS), and then re-suspended in PBS containing 10 µg/mL 2',7'-dichlorofluorescin (DCFDA, Molecular Probes-Invitrogen, CA). A stock of DCFDA was prepared fresh on the day of the experiment by dissolving in dimethyl sulfoxide to 10 mg/mL. The cell-DCFDA suspensions were then incubated for 2 hours at 30°C with shaking, washed once with PBS and re-suspended in 1 mL of PBS for direct analysis by flow cytometry using an Image Stream ISX100 (Amnis, WA). The level of reactive oxygen species was measured based on the emission of oxidized DCFDA dye in the green channel (517–527 nm).

## Bioinformatics screen of *S. cerevisiae* ORFs

Whole-genome ORFs of *S. cerevisiae* were surveyed for UUC codons using the CUT Codon Utilization bioinformatics tool [30]. The CUT tool uses existing genome and transcript data for *S. cerevisiae* to quantitate the number of times a transcript uses an individual codon or a run of a particular codon (i.e., UUC or UUCUUC). CUT then determines whether or not the codon, or codon run, occurs more or less than the average genome frequency. Statistically significant codon usage was determined by Z-score, with a cutoff of +1.96 representing the 95% confidence level, or an associated P-value of 0.05.

## Results

### 2'-O-ribose *trm* mutants are sensitive to oxidative stress

Considering the ability of oxidative stress to induce changes in 2'-O-ribose methylation events in *S. cerevisiae* [17, 18], together with observations of stress sensitivity phenotypes in other *S. cerevisiae trm* mutants (i.e., Trm9 and Trm4, respectively [13, 15]), we investigated whether 2'-O-ribose *trm* mutants are also sensitive to oxidative stress. Four 2'-O-ribose methyltransferases were identified in the *Saccharomyces* Genome Database (SGD), *TRM3*, *7*, *13*, and *44* ([39]Fig 1). Quantitative real-time reverse transcription PCR (qRT-PCR) using *TRM*-specific Taq-Man® gene expression assays confirmed that the deletion mutants do not express their respective *TRMs* (S1 Fig). Interestingly, in the *trm3*, *7* and *44* mutants, alternate 2'-O-ribose *TRMs* are up-regulated to varying degrees relative to the BY4741 wild type strain, supporting that compensatory mechanisms may exist at the transcriptional level.

*S. cerevisiae* strains lacking the above methyltransferases were retrieved from a haploid single-gene deletion library [32] and grown on plates containing hydrogen peroxide, rotenone, and acetic acid to induce oxidative stress. We found that all 2'-O-ribose methyltransferase mutants were sensitive to these toxicants relative to the wild type (control) BY4741 strain (Fig 2A). Notably, this sensitivity phenotype was most apparent on plates supplemented with hydrogen peroxide and was greatest for the *trm3* mutant (Fig 2A).

These plate-based assays represent a chronic exposure setting in which it can be challenging to differentiate the effect of the toxicant from the impact of the gene deletion on growth if any. Indeed, a slow-growth phenotype has been reported for *trm7* mutants [27, 28], and this phenotype was also detected here for the *trm7* mutant strain, while the growth of the *trm3*, *13*, *and 44* mutant strains was intermediate between *trm7* and wt BY4741 cells (S2 Fig). To further assess the oxidative stress sensitivity of the 2'-O-ribose *trm* mutants, we performed two acute exposure assays in which cells are normalized with respect to cell number prior to exposure. The resulting data reflect survival and recovery post-exposure rather than continued growth in the presence of the toxicant.

First, the effects of exposure to hydrogen peroxide, acetic acid and rotenone on cell death was determined by trypan blue staining to distinguish non-viable cells (i.e., positive for trypan blue) from viable cells. An eight-hour exposure to hydrogen peroxide produced high percentages of non-viable cells for the *trm3* and *trm7* mutant strains (Fig 2B, top panel). Exposure to rotenone for eight-hours significantly increased cell death across all strains, with the wild type BY4741 strain being largely refractory to this toxicant, which disrupts mitochondrial function to increase intracellular ROS (Fig 2B, middle panel). A significant increase in cell death was only observed for the *trm7* strain after an eight-hour exposure to acetic acid (Fig 2B, bottom panel). Overall, using viability assays as a measure of oxidative stress sensitivity, *trm7* mutants were the most sensitive to all toxicants, followed by *trm3* mutants (sensitive to hydrogen peroxide and rotenone), whereas *trm13* and *trm44* mutants were only sensitive to rotenone.

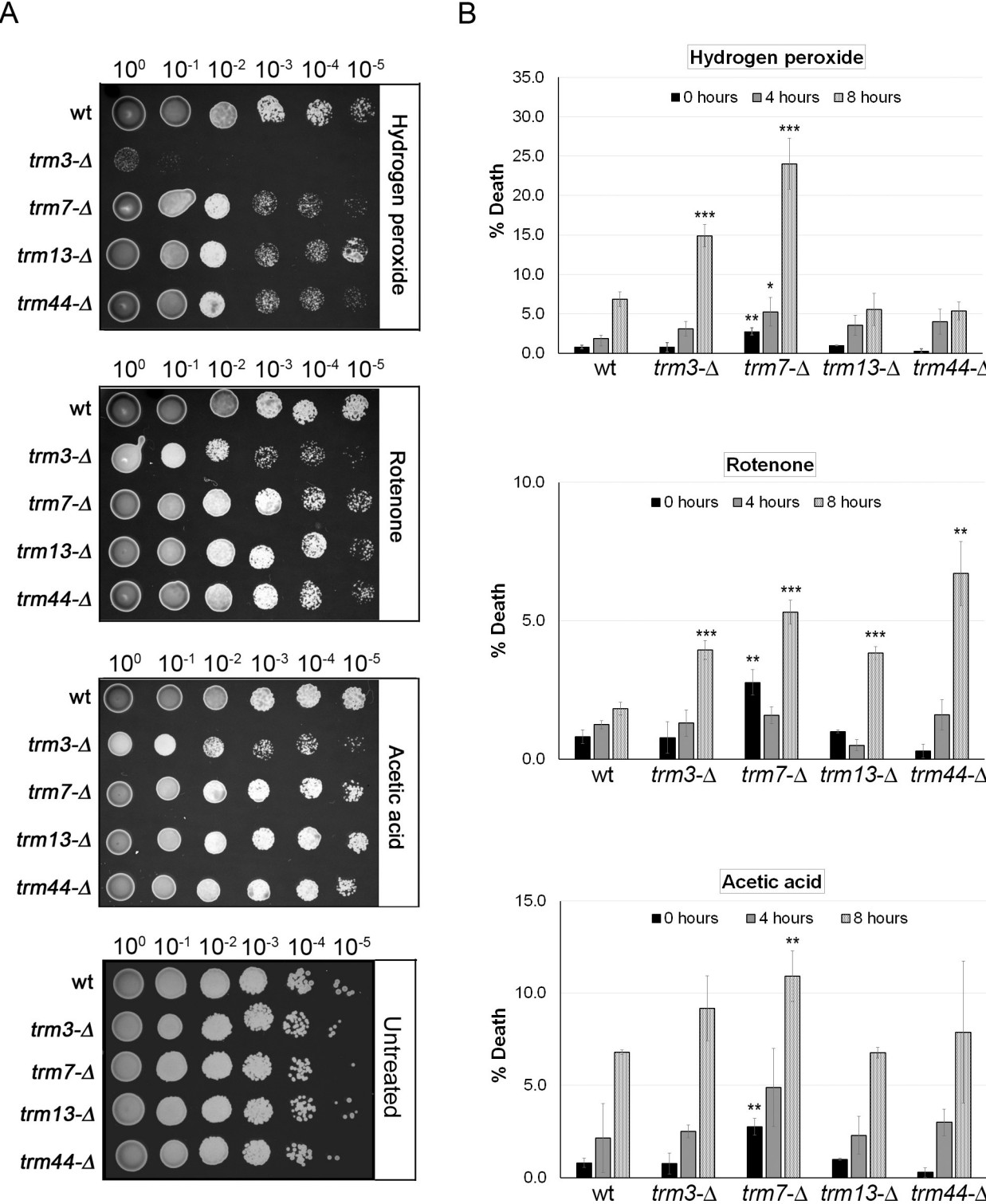

**Fig 2. Growth and survival of 2'-O-ribose *trm* mutants is inhibited in the presence of oxidative stress-inducing agents. A)** Wild type (wt) BY4741 and *trm* mutant strains were grown from a single colony overnight and then serially diluted in water as indicated. Five μl of each dilution was then plated on YPD agar plates containing acetic acid (20 mM), rotenone (20 μM), and hydrogen peroxide (2 mM), and the cells were grown for 3 days at 30°C. **B)** Cells were treated with hydrogen peroxide (40 mM), rotenone (400 μM), and acetic acid (80 mM) and non-viable cells were identified by trypan blue staining four and eight hours post-exposure. Untreated cells (i.e., zero hours post-exposure) served as the control. Cell death is shown as the mean number of trypan blue positive cells divided by total cells (i.e., trypan blue positive plus negative), expressed as a

percent, with error bars representing the standard deviation of the mean (n = 3). Significant differences in cell death percentages were determined using the t-test: *trm* mutants versus wild type under the same treatment condition and at the same time post-exposure (n = 3, *P < 0.05, **P < 0.01, ***P < 0.001).

Second, we used colony-forming assays to assess the ability of the 2'-O-ribose *trm* mutants to recover from an acute toxicant dose, sub-lethal to the wild type BY4741 strain. In this assay, all colonies formed post-exposure were counted, and the data indicate an ability to recover from toxicant exposure. We found that after acute exposure to hydrogen peroxide colony formation was significantly decreased for all 2'-O-ribose *trm* mutants compared to wild type cells (Fig 3A). Colony formation was similarly decreased after acute exposure to acetic acid and rotenone in *trm3* and *trm13* mutants (Fig 3A).

## Reintroduction of 2'-O-ribose methyltransferases attenuates sensitivity to oxidative stress

Since the 2'-O-ribose *trm* mutants were most sensitive to hydrogen peroxide, we used this toxicant to assess colony formation after introducing the open reading frames (ORFs) of the *TRM* genes under the control of a galactose (GAL) inducible promoter (i.e., *trm-Δ + TRM-ORF*). ORF constructs were available from a yeast open reading frame collection [33] for *TRM3*, *7*, and *13*, and their expression was selected for by initial growth in the absence of uracil. Under non-inducible growth conditions for ORF gene expression (i.e., + Dex: *TRM* "off") the colony-forming ability of the 2'-O-ribose mutants was reduced after exposure to hydrogen peroxide as expected. However, under growth conditions that induce ORF gene expression (i.e., + Gal: *TRM* "on") the number of colonies that survived hydrogen peroxide exposure significantly increased (Fig 3B). Although glucose is the preferred carbon source for *S. cerevisiae*, under +Gal-untreated conditions, the *trm3* and *trm7* mutants produced more colonies relative to the +Dex-untreated (results not shown), suggesting that some reprogramming occurs during the carbon source switch; thus, the data is presented relative to the same growth conditions concerning the carbon source. These results demonstrate some degree of rescue of oxidative stress sensitivity phenotypes by introducing 2'-O-ribose methyltransferase genes into their respective deletion mutants.

## Oxidative stress-induced nucleoside modification profiles in 2'-O-ribose *trm* mutants

We used direct infusion electrospray ionization mass spectrometry (ESI-MS) to quantitate the levels of ribonucleoside modifications in wild type BY4741 cells and 2'-O-ribose *trm* mutants before and after exposure to hydrogen peroxide, focusing on Am, Um, Cm and Gm. Exposure-induced changes are shown relative to unexposed wild type cells (Fig 4), which were calculated from discrete values determined using the abundance versus proxy method. This method measures the absolute signal intensity of the modified ribonucleoside over the sum of the absolute intensities of the canonical bases, taking into account any changes unmodified nucleosides [36].

We detected increases in Um, Cm, and Gm in wild type cells after exposure to hydrogen peroxide. These toxicant-induced increases in Um, Cm, and Gm were attenuated across all 2'-O-ribose *trm* mutants. Notably, in the cases of *trm3*, and *13*, this lack of induction occurred on a background of higher levels of Um, Cm and Gm, relative to wild type cells. Am was also higher in *trm13* mutants. These interesting (and unexpected) results lead us to question whether there is an oxidative stress phenotype in 2'-O-ribose *trm* mutants prior to exposure.

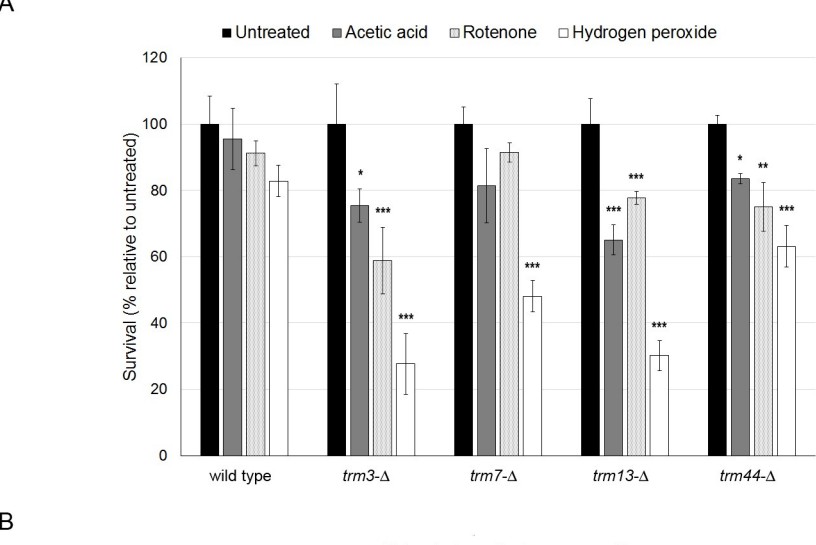

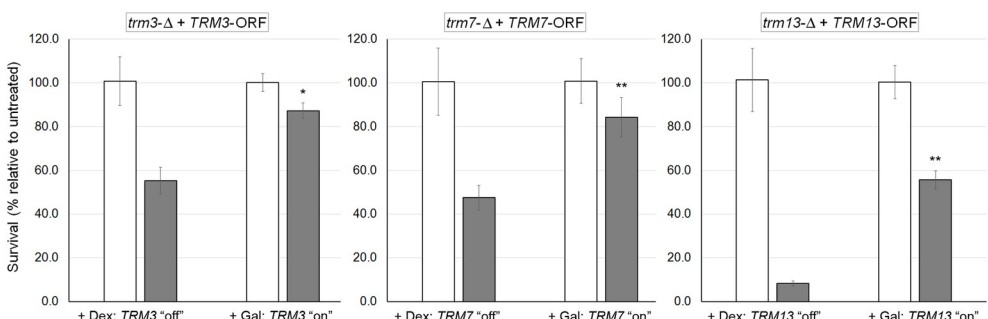

**Fig 3. Colony survival is decreased in 2'-O-ribose *trm* mutants after exposure to oxidative stress-inducing agents.
A)** Wild type (BY4741), *trm3*, *trm7*, *trm13*, and *trm44* mutant strains were grown to stationary-phase and exposed to acetic acid (20 mM), rotenone (100 μM), and hydrogen peroxide (10 mM). One hour post-exposure, the cultures were serially by a factor of $10^{-5}$, plated (100 μl), and the number of colonies was recorded after growth at 30°C for three days. Percent colony survival is shown as the mean for the toxicant relative to each individual strain under untreated conditions; error bars are standard deviations (n = 6 for wild type and n = 3 for *trm* mutants). Significant differences in colony formation and survival were determined using the t-test: *trm* mutants versus wild type under the same treatment condition ($^*P < 0.05$, $^{**}P < 0.01$, $^{***}P < 0.001$). **B)** *trm* mutants strains (as indicated) were transformed with their corresponding galactose (Gal)-inducible open reading frames (+ *TRM-ORF*). Cells grown to stationary phase in selection media (-Ura) + raffinose (carbon source) were diluted (3 fold) in media (-Ura) containing either galactose (+ Gal) or dextrose (+ Dex), and allowed further growth to stationary phase. Then, cells were split into separate tubes for one-hour control (water) or hydrogen peroxide (20 mM) treatments at 30°C, and similarly diluted (i.e., $10^{-5}$), plated, with colonies recorded three days later. Percent colony survival is shown as the mean ± standard deviations (*TRM3* and *TRM13*, n = 3; *TRM7*, n = 4). Significant differences in colony survival were determined using a Student t-test: cells exposed to hydrogen peroxide, +Gal (TRM "on") versus +Dex (TRM "off", $^*P < 0.005$, $^{**}P < 0.001$).

We investigated this possibility by staining cells with the 2',7'-dichlorofluorescein (DCFDA), a dye that oxidizes in the presence of intracellular reactive oxygen species (ROS), emitting fluorescence that is detected in the "green" channel using flow cytometry. The mean DCFDA intensity was significantly increased in all 2'-O-ribose *trm* mutants, with the *trm3* and *trm13* strains showing the most substantial increases compared to wild type cells (Fig 5). These preliminary results support that some redox mechanism(s) is deregulated in the *trm* mutants and are not indicative of the type of reactive oxygen species elevated, or hydrogen peroxide levels, which will be determined in future.

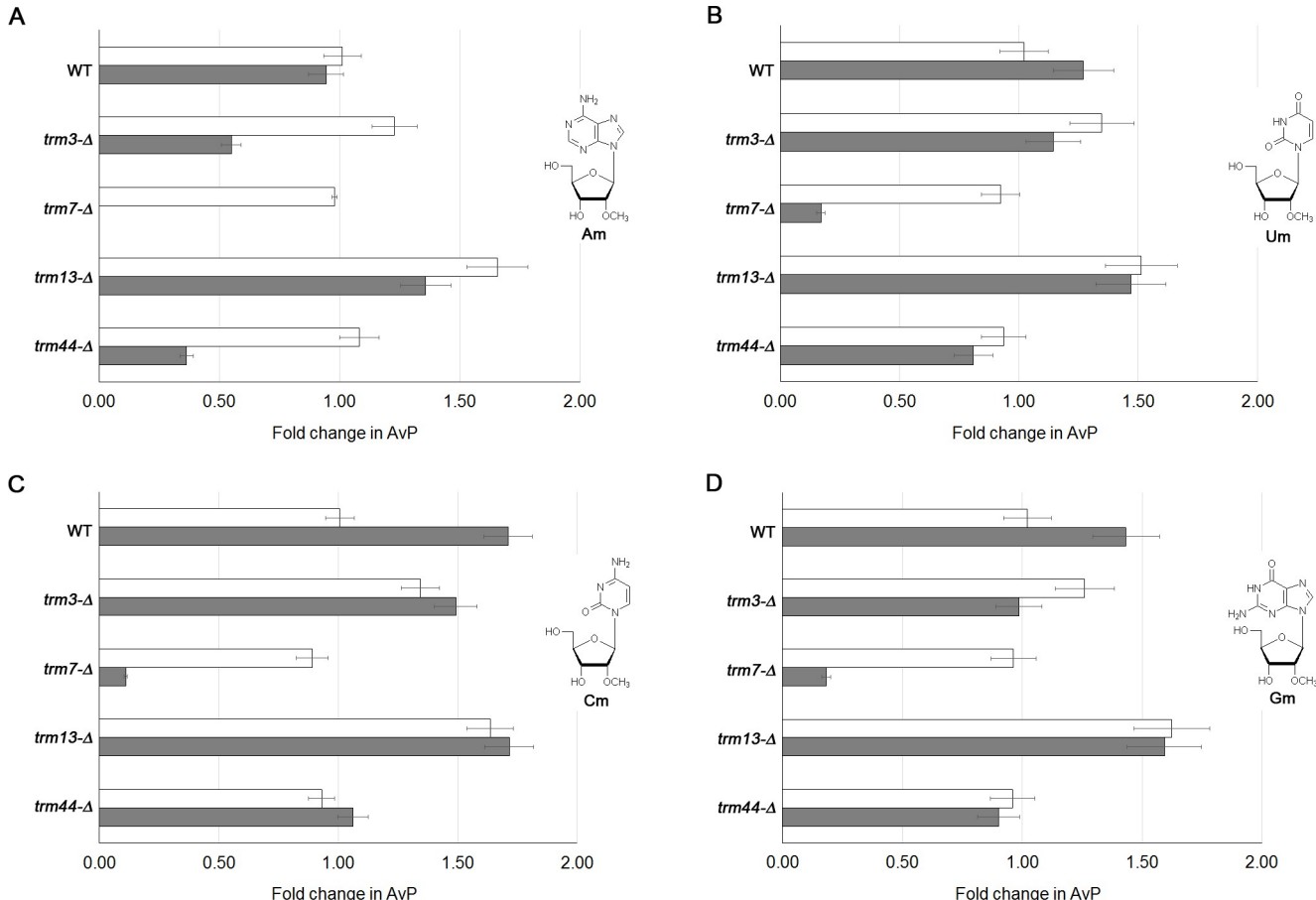

**Fig 4. Analysis of 2'-O-ribose methylation in *S. cerevisiae* exposed to hydrogen peroxide.** Wild type (wt) BY4741 and *trm* strains were cultured to log-phase growth (OD$_{600nm}$ = 0.6) and either left untreated (open bars) or exposed to hydrogen peroxide (20 mM for 1 hour, grey-filled bars). Ribonucleoside modifications were identified by direct infusion electrospray ionization mass spectrometry (ESI-MS). Quantitative measurements of 2'-O-methyladenosine (Am, panel A), 2'-O-methyluridine (Um, panel B), 2'-O-methylcytidine (Cm, panel C), and 2'-O-methylguanosine (Gm, panel D) were made using the abundance versus proxy method (AvP), defined as the absolute signal intensity over the sum of the absolute intensities of the canonical bases. Fold changes represent increases or decreases in AvP values relative to wt untreated cells and were calculated using the AvP values from S1 and S2 Tables, with error bars indicating the standard error of the mean change (n = 3).

In the case of *trm7* mutants, 2'-O-ribonucleosides were not induced by exposure to hydrogen peroxide (Fig 4). In fact, in cells treated with hydrogen peroxide, Um, Cm and Gm were decreased, and Am was undetectable. Overall, these results show a complex profile of 2'-O-ribose methylation events in *S. cerevisiae* exposed to hydrogen peroxide and suggest that some of these modifications act as part of both the oxidative stress response and in ROS maintenance under normal growth conditions. This complexity is further reflected in our global analysis of ribonucleotide modifications that are changed in the 2'-O-ribose *trm* mutants under both normal growth conditions and in response to hydrogen peroxide treatment (S1 and S2 Tables).

## Identification of candidate transcripts influenced by 2'-O-ribose methylation

Considering the oxidative stress sensitivity of the 2'-O-ribose *trm* mutants, we were interested in identifying transcripts that are translationally regulated by this type of methylation event.

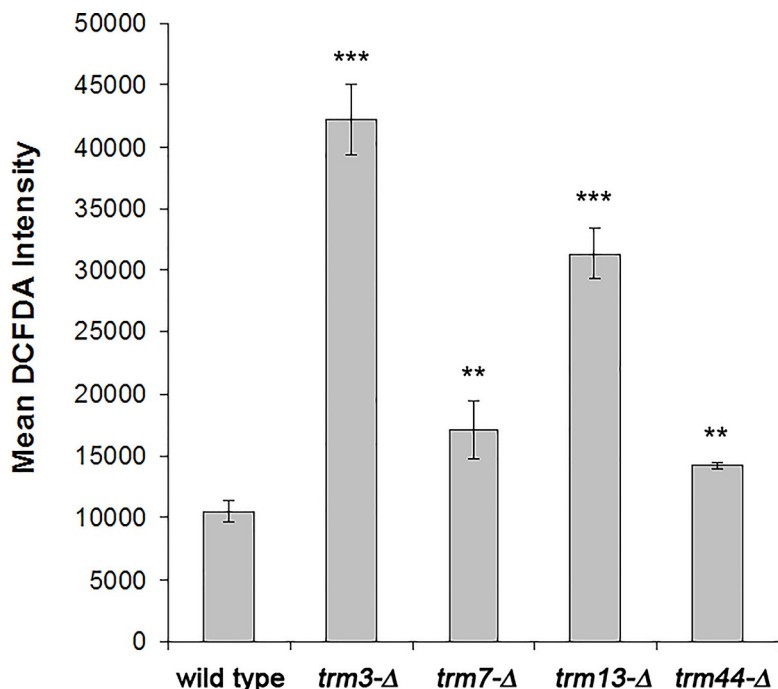

**Fig 5. 2'-O-ribose *trm* mutants have elevated levels of intracellular reactive oxygen species (ROS).** The indicated *trm* mutant strains were cultured to log-phase growth ($OD_{600nm} = 0.6$) and stained with 2',7'-dichlorofluorescin (DCFDA) to measure intracellular ROS. The fluorescence intensity emitted by oxidized DCFDA at 517–527 nm was determined by flow cytometry ($\geq$ 4000 cells per population), and is shown as the mean $\pm$ standard deviation (n = 4). Significant differences in mean fluorescence intensities were determined using the t-test: *trm* mutants versus wild type under similar treatment conditions (**P < 0.01, ***P < 0.001).

Based on previous work, we expect that stress-associated methylation of tRNAs enhances the translation of stress-responsive gene transcripts [40]. If this is true, then genes involved in the stress response will use codons whose translation is influenced by methylated tRNAs, including those methylated at the 2'-O-ribose position. We tested this idea using tRNA^Phe(GAA) as a model since it is a biologically relevant target of Trm7, [27, 28] and is methylated at the wobble position (underlined), a site that is known to influence codon selection (reviewed in)[5]. In addition, phenylalanine decoding represents a straightforward "two-choice" decision compared to other degenerate codons since it is decoded by either UUU or UUC, and *S. cerevisiae* Trm7-modified tRNA^Phe(GmAA) is thought to promote cognate UUC base-pairing interactions that favor the translation of UUC over UUU [41].

To identify transcripts that may be influenced by Trm7-mediated tRNA^Phe(GmAA) modifications, we took a bioinformatics approach using the gene-specific Codon UTilization (CUT) tool [30]. This tool identified a set of transcripts that use UUC to decode phenylalanine at a higher frequency when compared to all other transcripts of the *S. cerevisiae* genome. Moreover, these UUC-enriched transcripts correspond to genes that are involved biological processes associated with the oxidative stress response, as well as others that are involved in processes that regulate cell growth and metabolism (Table 1). Interestingly, these transcripts not only use UUC at a higher frequency compared to other transcripts, they also have UUC in duplicate or triplicate codon "runs", which is a rare event considering that there are only 24 reported open reading frames in *S. cerevisiae* that contain a UUCUUCUUC run, representing only 0.36% of the coding genome for this organism (S3 Table) [30].

**Table 1. Transcript open reading frames enriched in UUC codons for phenylalanine decoding.**

| Gene | Codon Sequence | # | Z-score* | Gene Ontology Term(s) | Biological Process |
|------|----------------|---|----------|-----------------------|--------------------|
| *CRT10* | UUC | 23 | 1.98 | | |
| | UUCUUC | 2 | 3.53 | RNR Transcription regulators | |
| | UUCUUCUUC | 1 | | | Stress Response |
| *YGK3* | UUC | 7 | -0.17 | | |
| | UUCUUC | 2 | 3.53 | Yeast homolog of GSK-3 | |
| | UUCUUCUUC | 1 | | | |
| *HIR3* | UUC | 38 | 3.99 | Positive regulation of histone acetylation; | |
| | UUCUUC | 3 | 5.48 | negative regulation of transcription | Chromatin Dynamics |
| | UUCUUCUUC | 1 | | involved in G1/S transition of mitotic cell cycle | |
| *ACC1* | UUC | 58 | 6.66 | | |
| | UUCUUC | 3 | 5.48 | protein import into nucleus; | |
| | UUCUUCUUC | 1 | | long-chain fatty acid biosynthetic process | Metabolism |
| *QNS1* | UUC | 13 | 0.64 | | |
| | UUCUUC | 2 | 3.53 | NAD biosynthetic process | |
| | UUCUUCUUC | 1 | | | |
| *FRE5* | UUC | 23 | 1.98 | | |
| | UUCUUC | 2 | 3.53 | Ferric-chelate reductase activity | Redox |
| | UUCUUCUUC | 1 | | | |
| *GNP1* | UUC | 20 | 1.58 | | |
| | UUCUUC | 2 | 3.53 | Amino acid and transmembrane transport | |
| | UUCUUCUUC | 1 | | | |
| *HXT2* | UUC | 29 | 2.78 | | |
| | UUCUUC | 5 | 9.38 | Glucose transport | Nutrient Acquisition |
| | UUCUUCUUC | 1 | | | |
| *HXT4* | UUC | 27 | 2.51 | | |
| | UUCUUC | 3 | 5.48 | Hexose transport | |
| | UUCUUCUUC | 1 | | | |

*Measures whether a transcript uses a certain codon sequence more (+) or less (-) than the global frequency.

### 2'-O-ribose translational targets are oxidative stress-sensitive

Our bioinformatics approach identified transcripts that are potential targets of Trm7-mediated 2'-O-ribose methylation, providing support for the idea that UUC-enriched transcripts rely on 2'-O-ribose methylation to enhance their translation in response to stress. As a first step towards testing this theory, we retrieved *S. cerevisiae* strains deficient for a subset of UUC-enriched genes (i.e., *CRT10*, *FRE5*, *HXT2*, *HIR3*, and *GNP1*) from our library of deletion strain mutants, and assessed colony survival after exposure hydrogen peroxide. We found significant sensitivity, to varying degrees, across all of the UUC-enriched gene mutants investigated thus far (Fig 6). These findings support that UUC-enriched gene transcripts encode proteins that play a functional role in recovering from oxidative stress and point to a potential role for Trm7-mediated regulatory effects at the translational level, which fits into our broader hypothetical model of how 2'-O-ribose methylation contributes to the oxidative stress response (Fig 7).

### Discussion

Here we have shown that the deletion of 2'-O-ribose methyltransferase genes in *S. cerevisiae* leads to sensitivity to oxidative stress-inducing toxicants. The degree of individual 2'-O-ribose

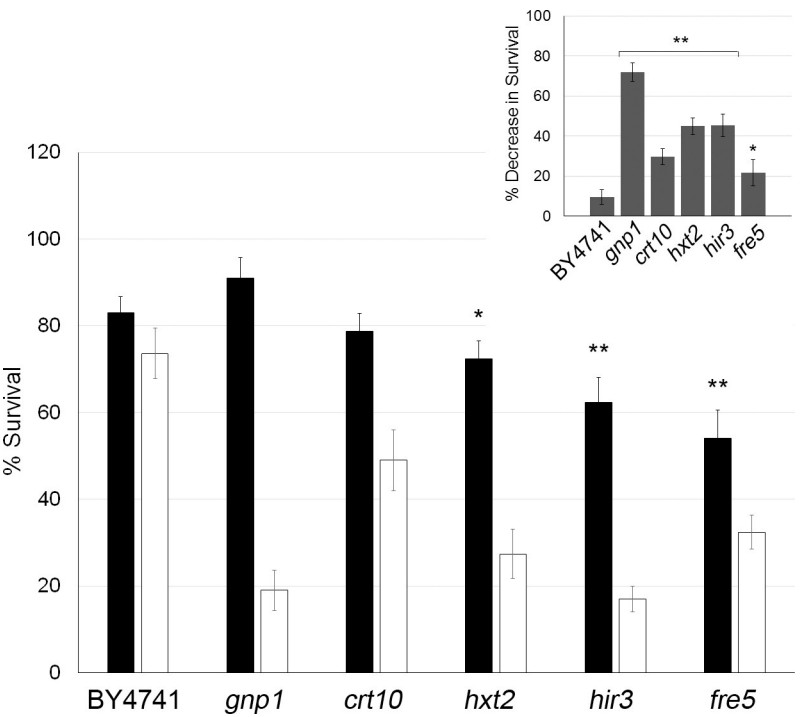

**Fig 6. Colony survival is decreased in UUC-enriched deletion mutants after exposure to oxidative stress-inducing agents.** Wild type (BY4741), *crt10*, *fre5*, *hxt2*, *hir3*, and *gnp1* mutant strains were grown to stationary-phase (i.e., $OD_{600}$ = 3.0) and exposed to hydrogen peroxide (10 mM) to induce oxidative stress. After 1 hour of exposure, the cultures were serially diluted to $9 \times 10^2$ cells per ml, plated (100 μl), and the number of colonies was recorded after growth at 30˚C for three days. In the central figure, percent survival is shown relative to untreated wt. The inset shows the magnitude of the decrease in survival (i.e., percent survival untreated minus treated conditions for individual strains). Significant differences in colony formation and survival were determined using a Student *t*-test: UUC-enriched mutants versus wild type under similar treatment conditions (n = 5, *P < 0.002, **P < 0.001).

*trm* mutant sensitivity was dependent upon assay endpoint (i.e., cell viability or cell survival to colony formation), and hydrogen peroxide produced the highest degree of oxidative stress sensitivity by colony assay across all *trm* mutants. Notably, although rotenone significantly increased cell death across all *trm* mutants, mitochondrial dysfunction may also be a contributing factor to a decrease in viability.

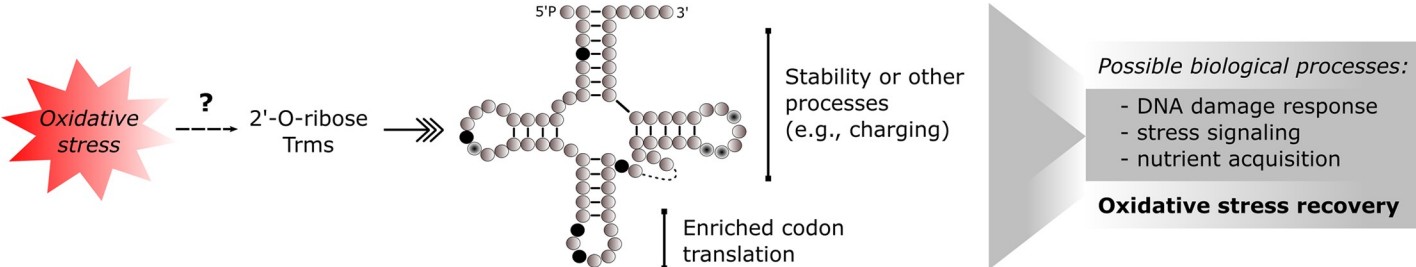

**Fig 7. A hypothetical model for how 2'-O-ribose tRNA methylation promotes oxidative stress recovery.** In this model, 2'-O-ribose Trms are responsive to oxidative stress exposure by unknown signaling events or pathways as indicated by the question mark. Trms then target various tRNAs for 2'-O-ribose methylation at positions that have previously been implicated in increased tRNA stability, amino acid charging, and enhanced translation of transcripts enriched in certain codons. Possible biological processes that may be involved in oxidative stress recovery via this type of translational regulation were identified at a bioinformatic level. The precise role of Trm-dependent 2'-O-ribose methylation for enhanced decoding of transcripts associated with these biological processes has yet to be determined.

Our analysis of oxidative stress-induced nucleoside modifications in the *trm* mutants revealed a complex profile of 2'-O-ribose methylation events, raising new questions. For example, why do cells deficient in *TRMs 3* and *13* increase overall levels of 2'-O-ribose modified nucleosides while exhibiting no observable growth impairment? Alone, these mutants were not deficient in any single 2'-O-ribose methylation event, arguing for compensatory mechanisms of 2'-O-ribose methylation that are sufficient for growth, but insufficient to overcome oxidative stress. Compensatory mechanisms further suggest an underlying stress phenotype, which included elevated levels of intracellular ROS in the *trm* mutants. We postulate that to offset the destabilizing effects of this ROS on tRNA molecules, 2'-O-ribose *trm* mutants undergo widespread increases in 2'-O-ribose methylation. Further research is required to define the molecular mechanisms behind this ROS-associated phenotype.

The *trm3* mutant strain showed a high degree of sensitivity across all toxicants relative to other 2'-O-ribose mutants. Trm3 is responsible for the formation of 2'-O-methylguanosine (Gm) at position 18, which occurs in several tRNA species including isotypes of tRNA$^{Leu}$, tRNA$^{His}$, tRNA$^{Tyr}$, tRNA$^{Ser}$, and tRNA$^{Trp}$; in these isotypes, Gm18 is adjacent to guanosine at position 19 [26, 42]. Our modification analysis shows an increase in Gm after exposure to hydrogen peroxide that was absent in the *trm3* mutants, raising the possibility that Trm3-mediated Gm18 formation may be a component of the oxidative stress response. A caveat to this interpretation is that our modification data is not positional, and so we cannot attribute exposure-induced Gm solely to the activity of Trm3 at position 18. A likely role for Trm3-mediated Gm modification in response to oxidative stress is to enhance the stability of tRNA molecules after exposure to toxicants that elevate ROS. Gm at position 18 of the D loop interacts with Ψ at position 55 and, together with other D- and TΨ-loop interactions (i.e., G19-C56 and G53-m1A), acts to stabilize the three-dimensional core of the tRNA molecule [11, 24, 25]. A parallel for this exists in thermophilic organisms where 2'-O-ribose methylation and 2-thiolation of t54, acts as a thermal adaptation to increase tRNA stability [20–23, 43]. In either environment—oxidative stress or extreme temperature—tRNA molecules would be susceptible to denaturation, particularly in non-duplex regions, such as the D-loop where G18 resides. It follows that Trm3-mediated Gm modification could represent an oxidative stress adaption of protein synthesis.

Trm7 interacts with Trm732 and Trm734 for respective 2'-O-ribose methylations of C32 and N34 to form Cm32, and Gm34 in tRNA$^{Phe}$, Cm32 and Cm34 for tRNA$^{Trp}$, and ncm5Um for tRNA$^{Leu(UAA)}$ [27, 28, 44]. As these Trm7-dependent modifications occur in the anticodon stem loop (ASL), they can influence anticodon: codon interactions [5–7]. By extension, the oxidative stress sensitivity seen here for the *trm7* mutants may be due to impaired codon selection, which negatively impacts the translation of stress-responsive transcripts (modeled in Fig 7). Reports of tRNA$^{Phe}$ genomic sequences with an AAA anticodon are rare, and whether or not these tRNA$^{PheAAA}$ genes are transcribed is not clear [45]. So, eukaryotes seem to rely upon a single anticodon in tRNA$^{Phe}$, GAA, to decode both UUC (cognate at the wobble) and UUU (non-cognate at the wobble). Thus, Gm methylation as a mechanism for promoting cognate codon recognition could represent a critical mechanism for degenerate codon selection in certain cellular contexts, including oxidative stress.

Several UUC-biased genes were identified here, and their corresponding *trm* mutants were sensitive to oxidative stress. *gnp1* mutants were the most sensitive relative to other UUC-biased mutants—a notable result since *GNP1* encodes an amino acid transmembrane transport protein, and *trm7* mutants have recently been shown to activate the general amino acid control (GAAC) response pathway [31]. These observations suggest that impaired translation of *GNP1* in *trm7* mutants is a component of the GAAC response and contributes to the pronounced oxidative stress sensitivity phenotype of these mutants.

Trm13 is responsible 2'-O-ribose methylation of position 4 in at least three species of *S. cerevisiae* tRNAs: tRNA$^{Gly(GCC)}$-Cm$_4$, tRNA$^{Pro}$-Cm$_4$, and tRNA$^{His}$-Am$_4$ [26, 42, 46]. Despite the overall elevated levels of Am in the *trm13* mutants, we detected a small but significant attenuation of the Am signal as induced by hydrogen peroxide. Together with the *trm13* mutant stress sensitivity phenotype, this suggests that there may be a Trm13 → Am$_4$ component to the oxidative stress response. Position 4 resides within an acceptor stem region that is engaged base-pairing interactions with residue 69 to form a stable duplex region; thus, a role in structural stability is not predicted for 2'-O-methylation at this site [46]. Another possibility is that Trm3-mediated position 4 methylation influences other processes of translation, such as translocation or aminoacylation. Indeed, in *E. coli*, 2'-O-ribose methylation within the acceptor stem can negatively influence translation [47], and some aminoacyl tRNA synthetases rely on tRNA features within the acceptor stem or variable loop region to determine tRNA identity for charging [48–51].

*trm44* mutants demonstrated the least amount of oxidative stress-sensitivity relative to the other 2'-O-ribose *trm* mutants studied herein. Trm44 is responsible for Um 2'-O ribose methylation at position 44 [52], and although U occurs at position 44 in isotypes of tRNA$^{Cys}$, tRNA$^{Tyr}$, and tRNA$^{Leu}$, to date, the only tRNA species targeted by Trm44 identified in *S. cerevisiae* is cytoplasmic tRNA$^{Ser}$ [26, 53, 54]. Attenuated increases in Um modifications in response to hydrogen peroxide treatment were detected in the *trm44* mutants, suggesting a Trm44 → Um component to the oxidative stress response. Um44 modification is thought to increase tRNA stability by maximizing base stacking and base interactions that promote the C3'-endo form (preferred for stability in thermophiles) of RNA [55]. Another role for Trm44-mediated Um44 modification could be linked to charging of tRNA$^{Ser}$ by seryl-tRNA synthetases, which rely on features within the acceptor stem and variable loop regions for recognition [48–51]. Thus, both stability and accurate tRNA$^{Ser}$ charging could be affected in *trm44* mutants, producing sensitivity to oxidative stress.

While this study substantiates a role for 2'-O-ribose tRNA methylation in the oxidative stress response, the question of how oxidative stress induces *trms* to methylate tRNAs remains. At least one eukaryote Trm, human TRM9L, is targeted for serine phosphorylation by the ERK-RSK pathway, as induced by hydrogen peroxide [56]; thus, 2'-O-ribose Trms may be similarly targeted for activating phosphorylation events, either by this or other stress-responsive kinase pathways. This effect could be further augmented by the oxidative inactivation of dual-specificity phosphatases, which dephosphorylate protein phospho-serine -threonine, and -tyrosine, effects that are known to be induced by ROS [57]. It is also possible that some increases in 2'-O-ribose methylation events do not come from an increase in *trm* activity per se, but are the result of changes in the pools of tRNAs available for *trm*-specific methylation (e.g., through transcriptional up-regulation or increased turn-over of unmethylated tRNA species). Overall, the toxicants used in this study generate diverse intracellular ROS, capable of damaging tRNAs as well as other biomolecules. The non-helical regions of tRNAs may be particularly susceptible to ROS since they are more likely to undergo spontaneous hydrolysis relative to double-stranded helical regions due to the "in-line" geometry of the phosphate and adjacent 2'-OH moiety, which can act as an attacking nucleophile [58]. This type of geometry does not occur in double-stranded helical regions, which are well-conserved among tRNAs and rarely methylated [58, 59]. Further, methylation of the 2'-OH sugar favors a 3'-endo sugar pucker in the helical region of tRNAs, which prevents the type of in-line geometry that is susceptible to spontaneous hydrolysis [11, 19, 60]. Thus, we propose that oxidative stress-associated 2'-OH ribose methylation represents a dual strategy that both maintains the structural stability of tRNAs and exerts translational effects post-exposure. In some instances, this strategy may involve promoting the translation of stress-responsive transcripts with a codon-bias

(e.g., Trm7 → tRNA$^{Phe}$), while in other instances, it promotes accuracy in charging (e.g., Trm44 → tRNA$^{Ser}$). Both situations represent combined epitranscriptomic strategies for oxidative stress recovery.

## Supporting information

**S1 Fig. *TRM* expression in wildtype and mutant strains.**
(DOCX)

**S2 Fig. Unchallenged growth of 2'-O-ribose *trm* mutants.**
(DOCX)

**S1 Table. Post-transcriptional RNA modifications, unchallenged.**
(DOCX)

**S2 Table. Hydrogen peroxide-induced post-transcriptional modifications.**
(DOCX)

**S3 Table. UUC-enriched open reading frames (ORFs).**
(DOCX)

## Acknowledgments

First we would like to thank all members of the Department of Biology and Chemistry at SUNY Polytechnic Institute for their support in enabling this research. In addition, we are grateful to Dr. E. Phizicky (University of Rochester) for providing an independent *trm7* mutant strain with an inducible Trm7 ORF to validate our observations. Also, we thank Dr. T. J. Begley (University at Albany) for providing access to his Open Biosystems deletion strain library. Dr. M. Fasullo provided advice on general techniques and specific protocols for *S. cerevisiae* growth and transformation, and generously offered his time to critically review this manuscript.

## Author Contributions

**Conceptualization:** Lauren Endres, Frank Doyle, Taylor Rahn, Bethany Lee, Jessica Seaman, Daniele Fabris.

**Data curation:** Lauren Endres, Rebecca E. Rose, Frank Doyle, Taylor Rahn, Bethany Lee, William D. McIntyre.

**Formal analysis:** Lauren Endres, Rebecca E. Rose, Daniele Fabris.

**Investigation:** Lauren Endres, Rebecca E. Rose, Taylor Rahn, Bethany Lee, Daniele Fabris.

**Methodology:** Lauren Endres, Rebecca E. Rose, Taylor Rahn, Bethany Lee, William D. McIntyre, Daniele Fabris.

**Project administration:** Lauren Endres.

**Resources:** Lauren Endres, Daniele Fabris.

**Software:** Lauren Endres, Frank Doyle, Daniele Fabris.

**Supervision:** Lauren Endres, Daniele Fabris.

**Validation:** Lauren Endres.

**Visualization:** Lauren Endres, William D. McIntyre.

**Writing – original draft:** Lauren Endres, Jessica Seaman.

**Writing – review & editing:** Taylor Rahn, Bethany Lee, Jessica Seaman.

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
