## [Decision Letter · Decision Letter 0]

13 Nov 2019

PONE-D-19-29059

2'-O-ribose methylation of transfer RNA promotes recovery from oxidative stress in Saccharomyces cerevisiae

PLOS ONE

Dear Dr. Endres,

Thank you for submitting your manuscript to PLOS ONE. After careful consideration, we feel that it has merit but does not fully meet PLOS ONE’s publication criteria as it currently stands. Therefore, we invite you to submit a revised version of the manuscript that addresses the points raised during the review process.

The reviewers raise a series of valid points, most of which you should be able to address by a thorough revision of the manuscript as such. It would be advantageous if you could also perfomr the experimental controls suggested by one reviewer; if not possible please argue carefully what the limits of your data are and how this could potentially affect any of your conclusions.

We would appreciate receiving your revised manuscript by Dec 28 2019 11:59PM. To enhance the reproducibility of your results, we recommend that if applicable you deposit your laboratory protocols in protocols.io, where a protocol can be assigned its own identifier (DOI) such that it can be cited independently in the future. For instructions see: http://journals.plos.org/plosone/s/submission-guidelines#loc-laboratory-protocols

We look forward to receiving your revised manuscript.

Kind regards,

Thomas Preiss, PhD

Academic Editor

PLOS ONE

Journal Requirements:

Reviewers' comments:

Reviewer's Responses to Questions

**Comments to the Author**

1. Is the manuscript technically sound, and do the data support the conclusions?

Reviewer #1: Partly

Reviewer #2: Partly

2. Has the statistical analysis been performed appropriately and rigorously? 

Reviewer #1: Yes

Reviewer #2: No

3. Have the authors made all data underlying the findings in their manuscript fully available?

Reviewer #1: Yes

Reviewer #2: No

4. Is the manuscript presented in an intelligible fashion and written in standard English?

Reviewer #1: Yes

Reviewer #2: Yes

5. Review Comments to the Author

Reviewer #1: Endres et al. present a study on the role of selected yeast methyltransferases (Trms) in response to oxidative stress. They focus on trm 3, 7, 13 and 44 deletions in the context of hydrogen peroxide, rotenone and acetic acid treatment. They show impaired viability of these deletion mutants to oxidative stress as well as impaired relative increase of 2-0-Methylation (albeit the background methylation in analysed mutants is increased). Finally, they use bioinformatic approach to suggest that trm7 could play a role in tRNA-Phe(GGA) 2-0-Methylation, which can potentially influence translation of UUC codon-enriched mRNAs.

Overall, it is a well-executed study and well-written manuscript. However, I do have two major points that need to be addressed:

1. It is important to confirm deletions and overexpressions by western blot or other quantitative proteomic methods. This is especially relevant due to unexpected increase of 2-0-Methylation in trm deletion strains. One potential explanation is that deletion of one trm results in the increase of others.

2. Figure 7 is very speculative and does not summarise the study well. The statement that 'methylation event enhances translation of transcripts that preferentially use UUC..' is unfounded. The experiments showing trm7 involvement in regulation of translation of selected mRNAs are missing. As it stands, Fig. 7 is way too hypothetical as the major message of the paper is that deletion of trm genes influence response to oxidative stress.

Reviewer #2: This manuscript by Endres et al, investigates the role of tRNA methyl transferases (Trms) in S. cerevisiae oxidative stress response and reveales that several Trms are necessary for optimal survival in response to H2O2. Most of the survival data and changes in tRNA modifications in response to H2O2 in the Trm mutant strains is sound to demonstrate that tRNA modifications are affected and important in the oxidative stress response in S. cerevisiae.

However, there are some issues related primarily to the representation of data and statistical analysis that require revision, which are listed below.

1. Figure 2: the data from supplemental figure S1 should be incorporated and quantified in Figure 2 of the manuscript, as this is an important point in regards to previous discrepancies in the literature. Moreover, the representative figure of 2A top panel suggests that the trm3 mutant is less viable than wt.

2. The data of Figure 3B is not adequately represented. If the authors are going to statistically compare TRM-on vs TRM-off in response to H2O2 all data should be expressed relative to TRM-off-untreated. This will take into consideration any changes in survival that are elicited by TRM-on-untreated conditions. Further, ANOVA statistical analysis should be performed to compare all 4 experimental groups.

It is assumed that gray shaded bars represent H2O2 treatment, but this was not indicated in the figure legend.

3. It is assumed that Figure 4 represents averages of 3 independent experiments (according to Tables S2 and S3 in supplemental data). However, it is unclear how statistical analyses were performed to obtain p<0.05, as stated in the legend. Error bars should show the experimental variability.

Moreover, tables in supplemental data should list +/- SD to indicate variability between experimental replicates. It is unclear what the difference in statistical significance is between cyan and dark blue squares. Authors should review the legends of supplemental figures for more detail, grammatical errors, and reference to supplemental table 1 rather than S2.

4. The authors should acknowledge that measurements of DCFDA have several caveats. As illustrated here: https://www.ncbi.nlm.nih.gov/pubmed/22027063. If possible, it would be advised that another method is employed to demonstrate that different strains have an altered basal oxidative stress level, and/or use negative and positive controls for DCFDA measurements.

5. In Figure 6, the data of survival of untreated mutants should also be represented relative to wt untreated cells to show if there is significant loss of survival even in unchallenged conditions (as one would expect for genes that are of vital importance). Which of these genes are only important to stress response (ie H2O2 challenge) rather than overall survival? This could also inform selection of genes to follow up in point 6. below.

6. In Figure 6 the authors demonstrate that several identified genes with UUC enriched codon use are also necessary for survival in response to H2O2, however the direct link to Trms is not made experimentally. The authors should demonstrate that protein expression of at least some of these target proteins are increased in response to H2O2 and that this is abrogated in response to Trm7 loss.

7. The discussion nicely describes the role tRNA modifications may have in stress response. However, some redundant discussions of the data from the results section could be shortened.

Instead the authors should elaborate on potential mechanism by which Trms sense oxidative damage. It is not discussed how H2O2 induces Trms to methylate tRNAs. Is it known if Trms are transcriptionally increased in resonse to H2O2? Or are they directly modified at the protein level to increase Trm activity, such as direct oxidation of Trm cysteines, or in response to upstream response to redox modified signaling pathways (e.g. oxidized and inactivated phosphatase?). Moreover, would it be expected that Trms identified here, such as Trm7, more susceptible to these types of redox dependent regulation?

Alternatively, as discussed in the last paragraph, is it possible that tRNAs are degraded that are not methylated by Trm7, and hence the ratio of methylated to unmethylated tRNAs increases? In this case, is it possible that there is not net change in Trm7 activity in response to H2O2? This should be discussed.

6. PLOS authors have the option to publish the peer review history of their article (what does this mean?). If published, this will include your full peer review and any attached files.

Reviewer #1: No

Reviewer #2: No

---

## [Author Response · Author response to Decision Letter 0]

20 Jan 2020

Reviewer #1: Endres et al. present a study on the role of selected yeast methyltransferases (Trms) in response to oxidative stress. They focus on trm 3, 7, 13 and 44 deletions in the context of hydrogen peroxide, rotenone and acetic acid treatment. They show impaired viability of these deletion mutants to oxidative stress as well as impaired relative increase of 2-0-Methylation (albeit the background methylation in analysed mutants is increased). Finally, they use bioinformatic approach to suggest that trm7 could play a role in tRNA-Phe(GGA) 2-0-Methylation, which can potentially influence translation of UUC codon-enriched mRNAs.

Overall, it is a well-executed study and well-written manuscript. However, I do have two major points that need to be addressed:

1. It is important to confirm deletions and overexpressions by western blot or other quantitative proteomic methods. This is especially relevant due to unexpected increase of 2-0-Methylation in trm deletion strains. One potential explanation is that deletion of one trm results in the increase of others.

Response: We agree that TRM expression needed to be assessed in the mutant strains, which could provide relevant information concerning potential compensatory mechanisms. To address this shortcoming, we performed qRT-PCR on reverse-transcribed cDNAs from each trm deletion mutant with TRM-specific TaqMan gene expression assays. After comparing TRM expression in the mutants to that in the wildtype we have: i) confirmed that the deletion mutants do not express their respective Trms, and ii) identified several TRMs that are up-regulated the mutant strains, suggesting that compensatory mechanisms may exist, at least at the transcriptional level. This new data appears in S1 Fig., and is discussed in the results section.

 These results will inform proteomic studies, which are currently limited by the availability of commercial antibodies to our TRMs of interest (i.e., only available for TRM13 from Biorbyt). We would like to note that yeast strains with TAP-tagged open reading frames (ORFs) are widely used to study the yeast proteome using anti-TAP antibodies for western blotting, which could be a factor in the limited availability of antibodies for specific Trms in yeast. Further, the TAP-tagged strains could not be used here to address compensatory mechanisms as they do not harbor the relevant TRM deletion, having instead a TAP-tagged ORF. Our lab is not equipped for antibody production; however, as antibodies for Trms 3,7 and 44 become available, we aim to perform the suggested western blots.

2. Figure 7 is very speculative and does not summarise the study well. The statement that 'methylation event enhances translation of transcripts that preferentially use UUC..' is unfounded. The experiments showing trm7 involvement in regulation of translation of selected mRNAs are missing. As it stands, Fig. 7 is way too hypothetical as the major message of the paper is that deletion of trm genes influence response to oxidative stress.

Response: This figure has been completely revised to summarize the major message of the paper that 2'-O-ribose methylation acts as a response to oxidative stress. The biological processes involved in this response (as suggested by our bioinformatic screen) are now represented as purely hypothetical. In retrospect, the model for enhanced translation of UUC-enriched transcripts was designed to guide prospective studies, reflecting where we are going and not where we are in this paper. We thank the reviewer for this sound criticism.

Reviewer #2: This manuscript by Endres et al, investigates the role of tRNA methyl transferases (Trms) in S. cerevisiae oxidative stress response and reveales that several Trms are necessary for optimal survival in response to H2O2. Most of the survival data and changes in tRNA modifications in response to H2O2 in the Trm mutant strains is sound to demonstrate that tRNA modifications are affected and important in the oxidative stress response in S. cerevisiae.

However, there are some issues related primarily to the representation of data and statistical analysis that require revision, which are listed below.

1. Figure 2: the data from supplemental figure S1 should be incorporated and quantified in Figure 2 of the manuscript, as this is an important point in regards to previous discrepancies in the literature. Moreover, the representative figure of 2A top panel suggests that the trm3 mutant is less viable than wt.

Response: The previous Figure S1 is now incorporated into Figure 2. To quantitate the effects of TRM deletion on the growth of the individual mutant strains, we developed a new assay to monitor growth over time with an automated microplate reader. This quantitative growth-curve data appears in a new Figure S2, and confirms the previous reports of a slow-growth phenotype for trm7. Also, we show that the growth of trm3, 13 and 44 is also mildly impaired (i.e., being intermediate between wildtype and trm7). Together, we feel that this addresses discrepancies in the literature concerning the growth effects of trm3, and also strengthens work by adding new information about the growth phenotypes of trm13 and 44. 

2. The data of Figure 3B is not adequately represented. If the authors are going to statistically compare TRM-on vs TRM-off in response to H2O2 all data should be expressed relative to TRM-off-untreated. This will take into consideration any changes in survival that are elicited by TRM-on-untreated conditions. Further, ANOVA statistical analysis should be performed to compare all 4 experimental groups.

It is assumed that gray shaded bars represent H2O2 treatment, but this was not indicated in the figure legend.

Response: The absence of a figure legend indicating that the gray shaded bars are the H2O2 treated group was an unintentional omission. This figure is now corrected. For robust induction of the galactose-inducible TRM open reading frames, trm transformants were cultured first to stationary phase in raffinose (to deplete intercellular pools of glucose) followed by a switch to galactose to induce expression ("on") or dextrose to repress expression ("off"). Glucose is the preferred carbon source for S. cerevisiae; however, we noted that under gal-untreated conditions, the trm3 and trm7 mutants seemed to switch their preference to galactose, which was not the case for trm13. These results suggest that some reprogramming occurs during the carbon source switch that is not uniform among the trm mutants. Although this is an intriguing observation, it would require much further experimentation to explain taking us outside of the context of the experiment, which was to determine whether the reintroduction of TRM expression rescues H2O2-associated decreases in survival under similar growth conditions. We feel that the data as represented is best since it reflects the same growth conditions concerning the carbon source. Additional text has been added to the results section to make this point clear. 

3. It is assumed that Figure 4 represents averages of 3 independent experiments (according to Tables S2 and S3 in supplemental data). However, it is unclear how statistical analyses were performed to obtain p<0.05, as stated in the legend. Error bars should show the experimental variability.

Moreover, tables in supplemental data should list +/- SD to indicate variability between experimental replicates. It is unclear what the difference in statistical significance is between cyan and dark blue squares. Authors should review the legends of supplemental figures for more detail, grammatical errors, and reference to supplemental table 1 rather than S2.

Response: We determined standard error for the fold change data, which is now added to Figure 4 as error bars to show experimental variability. During this analysis, we identified an inadvertent duplication of the fold change data for trm44 untreated and have corrected the figure and revised the results and discussion sections accordingly. The data in Figure 4 were calculated using the AvP values from supplemental tables S3 and S4, and an explanation of the statistical analysis (i.e., 95% confidence) has been added to the table descriptions, referencing the methods section. 

Concerning the mean AvP values in tables S3 and S4, standard deviations (+) have been added. Also added is an explanation of the difference between the values appearing in the cyan and dark blue shaded cells, with an additional color code (red) to indicate post-translational modifications detectable in some strains but not in wt untreated cells. Please note that these tables have been shifted to accommodate new experimental data. Legends have been reviewed and corrected for grammatical errors.

4. The authors should acknowledge that measurements of DCFDA have several caveats. As illustrated here: https://www.ncbi.nlm.nih.gov/pubmed/22027063. If possible, it would be advised that another method is employed to demonstrate that different strains have an altered basal oxidative stress level, and/or use negative and positive controls for DCFDA measurements.

Response: Agree. We have added text describing the limitations of our observations with the DCFDA stained cells. Namely, that our results only support that some redox mechanism(s) is deregulated in the trm mutants. They do not indicate the type of reactive oxygen species elevated, or provide a direct measurement of intracellular hydrogen peroxide. We undertook the DCFDA staining to provide some explanation that accounts for the increase in oxidative stress-associated tRNA modifications under unchallenged conditions. As stated, this was an interesting and unexpected result, and we feel that the results were presented this way (i.e., as a preliminary and requiring future investigation to explain). We appreciate this point because discussing the limitation of the DCFDA results further emphasizes their preliminary status. In future, both the nature and source of the ROS need to be explored using specific probes and assays (e.g., dihydrorhodamine to measure peroxynitrite, 8-Isoprostane to detect lipid peroxidation, Mito-SOX to source mitochondrial superoxide, catalase activity to measure hydrogen peroxide, etc.), which constitute in an independent investigation.

5. In Figure 6, the data of survival of untreated mutants should also be represented relative to wt untreated cells to show if there is significant loss of survival even in unchallenged conditions (as one would expect for genes that are of vital importance). Which of these genes are only important to stress response (ie H2O2 challenge) rather than overall survival? This could also inform selection of genes to follow up in point 6. below.

Response: The data in Figure 6 no longer shows survival relative to untreated individual mutants. It is now presented as percent survival relative to untreated wt in which the mutants with a loss of survival under unchallenged conditions are apparent (i.e., hxt2, hir3, and fre5). To convey the effects of hydrogen peroxide on mutant colony survival (outside of the loss of survival due to the genetic mutation), we determined the magnitude of the decrease in survival. That is the percent decrease in survival; untreated minus treated conditions. This percent decrease in survival of the mutants post-exposure is compared to wt and presented as an inset to the original figure. 

6. In Figure 6 the authors demonstrate that several identified genes with UUC enriched codon use are also necessary for survival in response to H2O2, however the direct link to Trms is not made experimentally. The authors should demonstrate that protein expression of at least some of these target proteins are increased in response to H2O2 and that this is abrogated in response to Trm7 loss.

Response: We acknowledge that we have not made any direct links between the genes that are UUC-enriched and survival downstream of Trms. This point was also raised by reviewer 1. To address this problem, we have revised the model in Figure 7 to reflect the significant findings of the paper rather than our hypothetical model of Trm7-associated UUC-biased codon translation, which at this stage of our investigation is not supported by the data. 

 Assessing protein expression of the transcripts implicated as Trm7 targets is an essential step towards linking them to oxidative stress survival in our model. We are somewhat restricted by the availability of antibodies, finding only anti-Gnp1 (ThermoFisher) and -Hir3 (Abcam), which would need to be optimized and validated for detection in yeast. Instead, we intend to use yeast strains with TAP-tagged open reading frames (ORFs) to study protein expression and are currently optimizing a western blot protocol for use with a yeast anti-TAP antibody. The caveat to this approach is that each trm deletion mutation must be introduced into the TAP-tagged strains. At this stage, this research effort is ongoing, and considering that we presented the UUC-enriched transcripts as only candidate Trm7 targets, we think that their further characterization represents the next chapter of this story. 

7. The discussion nicely describes the role tRNA modifications may have in stress response. However, some redundant discussions of the data from the results section could be shortened.

Instead the authors should elaborate on potential mechanism by which Trms sense oxidative damage. It is not discussed how H2O2 induces Trms to methylate tRNAs. Is it known if Trms are transcriptionally increased in resonse to H2O2? Or are they directly modified at the protein level to increase Trm activity, such as direct oxidation of Trm cysteines, or in response to upstream response to redox modified signaling pathways (e.g. oxidized and inactivated phosphatase?). Moreover, would it be expected that Trms identified here, such as Trm7, more susceptible to these types of redox dependent regulation?

Response: These are excellent questions, and got us thinking about potential upstream events that activate Trm activity after hydrogen peroxide exposure. New ideas about potential mechanisms of Trm activation have now been added to the discussion and take the place of some sections that were redundant with the results (see below). Namely, we discuss the idea that at least one eukaryote Trm, human TRM9L, is targeted for serine phosphorylation by the ERK-RSK pathway, as induced by hydrogen peroxide (Gu et al., 2018); thus 2'-O-ribose Trms may be similarly targeted for activating phosphorylation events, either by this or other stress-responsive kinase pathways. We further speculate that this effect could be augmented by the oxidative inactivation of dual-specificity phosphatases, which dephosphorylate protein phospho-serine -threonine, and -tyrosine; effects that are known to be induced by ROS (Ostman, et al., 2011). 

Overall, our focus in this paper was to implicate 2'-O-ribose tRNA methylation in the oxidative stress response and discuss the biological importance of this type modification from two perspectives: i) its effects on the tRNA molecule itself, and ii) its potential ability to influence the translation of genes with the potential to act in oxidative stress repair/recovery. Thus, we have not concentrated on upstream activation mechanisms, but from this input, we plan to do so now. Accordingly, we have recently requested access to the TAP-tagged strains from our collaborator to fully characterize changes in both TRM transcript and Trm protein levels after hydrogen peroxide exposure.

Alternatively, as discussed in the last paragraph, is it possible that tRNAs are degraded that are not methylated by Trm7, and hence the ratio of methylated to unmethylated tRNAs increases? In this case, is it possible that there is not net change in Trm7 activity in response to H2O2? This should be discussed.

Response: Yes, unmethylated tRNAs could be degraded, affecting the methylated: unmethylated ratios. In fact, our model predicts that 2'-O-ribose methylation increases tRNA stability acting in concert with anti-codon modifications (e.g., as catalyzed by Trm7) to enhance translation. So, it is certainly possible that while Trm7 activity remains the same, the pools of tRNAs targeted by Trm7 are protected from degradation, increasing the methylated: unmethylated ratio. Also, there could be de novo transcriptional increases in Trm7 target tRNAs, not requiring an increase in enzyme activity to be methylated. This scenario would also increase the methylated: unmethylated ratio. As a first step to addressing these possibilities, a global assessment of tRNA transcript expression before and after hydrogen peroxide exposure is needed. We have revised the discussion to address these excellent points in more detail. 

Redundant sections removed or reworded to accommodate new discussion points:

- lines 446 to 449, "Um, Cm and Gm increased in wild type cells exposed to hydrogen peroxide, a response that was attenuated in trm3, 13, and 44 mutants; however, this attenuation occurred on a background of elevated Um, Cm and Gm before exposure, a curious result that raises new questions."

- lines 456 to 458, "Initial studies of trm3 mutants did not reveal a growth impairment (42), and we confirmed that the growth of trm3 mutants is comparable to that of the wild type strain, BY4741. However, when challenged with oxidative stress (chronic and acute),"

- lines 485 to 487, "Here we suggest that under conditions of oxidative stress, Trm7-dependent tRNAPhe(GmAA)) promotes UUC translation of mRNA transcripts with a role in the oxidative stress response (Fig 7)"

- lines 493 to 496, "There are conflicting reports on whether or not trm13 mutants have impaired growth (46, 47), and the growth of trm13 mutants here was comparable to wild type cells. However, upon toxicant exposure, the growth of trm13 mutants was impaired relative to wild type cells in both plate-based and colony-forming assays."

---

## [Decision Letter · Decision Letter 1]

30 Jan 2020

2'-O-ribose methylation of transfer RNA promotes recovery from oxidative stress in Saccharomyces cerevisiae

PONE-D-19-29059R1

Dear Dr. Endres,

We are pleased to inform you that your manuscript has been judged scientifically suitable for publication and will be formally accepted for publication once it complies with all outstanding technical requirements.

With kind regards,

Thomas Preiss, PhD

Academic Editor

PLOS ONE

Additional Editor Comments (optional):

Reviewers' comments:

Reviewer's Responses to Questions

**Comments to the Author**

1. If the authors have adequately addressed your comments raised in a previous round of review and you feel that this manuscript is now acceptable for publication, you may indicate that here to bypass the “Comments to the Author” section, enter your conflict of interest statement in the “Confidential to Editor” section, and submit your "Accept" recommendation.

Reviewer #1: All comments have been addressed

Reviewer #2: All comments have been addressed

2. Is the manuscript technically sound, and do the data support the conclusions?

Reviewer #1: Yes

Reviewer #2: Yes

3. Has the statistical analysis been performed appropriately and rigorously? 

Reviewer #1: Yes

Reviewer #2: I Don't Know

4. Have the authors made all data underlying the findings in their manuscript fully available?

Reviewer #1: Yes

Reviewer #2: Yes

5. Is the manuscript presented in an intelligible fashion and written in standard English?

Reviewer #1: Yes

Reviewer #2: Yes

6. Review Comments to the Author

Reviewer #1: All my comments have been addressed. I think it is an interesting study that deserves to be published.

Reviewer #2: (No Response)

7. PLOS authors have the option to publish the peer review history of their article (what does this mean?). If published, this will include your full peer review and any attached files.

Reviewer #1: No

Reviewer #2: No

---

## [Editor Report · Acceptance letter]

4 Feb 2020

PONE-D-19-29059R1 

2'-O-ribose methylation of transfer RNA promotes recovery from oxidative stress in *Saccharomyces cerevisiae*

Dear Dr. Endres:

I am pleased to inform you that your manuscript has been deemed suitable for publication in PLOS ONE. Congratulations! Your manuscript is now with our production department. 

With kind regards,

on behalf of

Prof Thomas Preiss 

Academic Editor

PLOS ONE